



# Are fire mediated feedbacks burning out of control?

Jon Lloyd[1,2] and Elmar Veenendaal[3]

[1]Department of Life Sciences, Imperial College London, SL5 7PY, Ascot, UK
[2]School of Tropical and Marine Sciences and Centre for Terrestrial Environmental and Sustainability Sciences, James Cook University, Cairns, 4870, Qld, Australia
[3]Plant Ecology and Nature Conservation Group, Wageningen University, 6700 AA Wageningen, The Netherlands

*Correspondence to:* Jon Lloyd (jonathan.lloyd@imperial.ac.uk)

**Abstract.** Contributing to recent discussions regarding the evidence for and against forest and savanna representing alternative steady states (ASS) across much of the tropical lands, we here address several issues raised by Staal and Flores (2015) in a recent commentary published in this journal (*Biogeosciences*, 12, 5563-5566, 2015). Our analysis shows that — in what could alternatively be titled "A Tale of Five Fallacies" -– substantial errors in reasoning exhibited by both Staal and Flores (2015) and

the ASS community in general in terms of arguments invoked as providing support for ASS as a key factor determining tropical vegetation distributions. Specifically we: (1) demonstrate that bimodal distributions of canopy cover need not necessarily be associated with ASS ("fallacy of confirmation bias"); (2) show that models suggesting the mathematical feasibility of ASS can never be taken as any sort of proof as to their actual existence ("fallacy of misplaced concreteness"); (3) conclude that studies failing to make climate-independent associations between soil properties and forest/savanna distributions (and thereby concluding that ASS must be present) have inevitably failed to measure the right things. Or in many cases, not even made

the required measurements ("fallacy of suppressed evidence"). Moreover, we also find: (4) assertions that ASS associated concepts such as detrimental effects of fire on soil chemical properties and positive effects of forest tree species on soil fertility are totally without foundation ("fallacy of wishful thinking") and (5) a strong tendency amongst ASS advocates to take results from temperate ecosystems and incorrectly assume that this provides support for the existence of ASS in the tropical forest and

savanna lands ("fallacy of hasty generalisation"). We conclude all arguments presented to date in support of the widespread existence of ASS in the tropical regions to be flawed. As an alternative, we suggest that forest-savanna transitions may be better understood as reflecting the effects of soil physical and chemical properties on tropical vegetation structure and function with fire-effected feedbacks simply serving to reinforce these patterns through a "sharpening switch" mechanism.

## 1  Introduction

In a recent commentary in this journal, Staal and Flores (2015) (here on referred to as "SaF") have questioned some key parts of a recent analysis of the structure of over 60 forest and savanna plots sampled in Africa, Australia and South America as undertaken by Veenendaal et al. (2015). In the latter we specifically noted that "the transitional vegetation formations described in our study do not present a spatially complete frequency distribution of all savanna and forest formations present across the planet" and analysed our data in such a way that this inevitable lack of true global representativeness did not unnecessarily





confound our results. This was achieved by examining how changes in the relative abundance of the different component strata and/or lifeforms related to changes on total canopy cover across the forest/savanna transition. One key conclusion from that paper was that, once the usually neglected subordinate woody layer was taken into account, then differences in total canopy cover between adjacent forest and savanna stands were much less than the (apparent) difference observed when only the upper

strata were considered Amongst other things, Veenendaal et al (2015) suggested that recent papers by Hirota et al. (2011) and Staver et al. (2011a) may have overestimated the extent of an apparent disjunction in canopy cover around values of 0.6 that exists in the global dataset of Hansen et al. (2005) as derived from remote sensing products; this potentially leading to spurious claims of a verification of the existence of forest and savanna as Alternative Steady States (ASS). According to the ASS theory, it is possible for more than one distinct vegetation formation type to exist under the same set of climatic/edaphic conditions.

With a "stable state" defined as the condition to which an ecosystem (or in its original construct a "community") will always return after any small perturbation (May, 1977) one excellent illustrative example of the ASS concept – also cited by Staal and Flores (2015) in support of their arguments – is as first presented for moist temperate forest and open eucalyptus woodland types in Tasmania (Australia) by Jackson (1968). Labelling this process "ecological drift", Jackson argued that should – by chance – a repeated series of crown–replacing fires occur for any given moist forest, then it could undergo a gradual transition to

one of several potential "degraded scrub forest" states which are dominated by fires. The latter burn regularly due to their open nature allowing the presence of flammable grasses. Jackson then argued that as forest species are essentially fire–intolerant (requiring a long fire–free period for their establishment) this meant that after the transformation to degraded scrub forest, it was very unlikely (though not impossible) for the scrub–forest community to return to it's much higher biomass moist forest state. Jackson also argued that this process should be more likely to occur on low–nutrient sandy soils on a resistant quartzitic

substratum.

Despite being almost entirely conjectural, there is little doubt that Jackson et al. (1968) remains a landmark paper of highly original scientific thought. But – contrary to what is claimed by SaF about this work – being essentially a data-free concepts paper, it hardly "demonstrates a feedback between low tree cover and fire". And indeed, as we show below here, there are many similar statements in SaF where conjecture is mistakenly taken as "proof". Moreover, a failure to make the appropriate distinc-

tion applies to many arguments advanced in support of ASS not only by SaF and also by many of those who it would seem believe in the widespread existence of ASS (see for example Scheffer et al 2015): this seemingly arising from a fundamental misconception about what constitutes "evidence" for a theory and what does not.

There are quite a few lines of argument presented by SaF, but – as detailed below –all of these are questionable at best. To help divide our analysis of the actual evidence pointing to the existence for ASS as a key determinant of tropical vegetation

types – which by necessity extends beyond SaF's specific comments to the wider literature they cite – we categorise the quite numerous questionable interpretations of phenomena having been taken as providing support for ASS into five fallacious categories; a fallacy being defined here quite simply as an "error in reasoning"(Woods, 2013).





## 2   Bimodal distributions, ASS and the "fallacy of confirmation bias"

Writing their commentary on Veenendaal et al. (2015), SaF suggested that because the frequency distribution of canopy cover in that limited dataset was bimodal (even with shrubs included in the calculation) that this shows that the MODIS tree–cover product "rightly captures bimodality". But, irrespective of the fidelity of the Hansen et al. (2005) data set and general questions

concerning its applicability for vegetation distributional analyses as raised for example by Hanan et al. (2014) and Veenendaal et al. (2015), it should be obvious that analyses such as have been done by Hirota et al. (2011) and Staver et al. (2011a) are only possible in the first place because of their global coverage removing any concerns of lack of representativeness due to small sample size. And, as noted already, because our sampling extent was not sufficient to allow us to suppose any approximation of a general representativeness of the true statistical distribution of fractional canopy cover we specifically resisted any temptation

to undertake analyses such as done by Hirota et al. (2011) and Staver et al. (2011a). Indeed, SaF also seem to recognise this very problem stating that "the field plots in Veenendaal et al. (2015) are not randomly selected from all possible tropical forest-savanna ecotones". But then curiously, despite their apparently also realising our dataset was clearly not appropriate for any sort of analysis as regards the relative distributions of crown cover classes, they then go on to do it anyway (!). Indeed, all their Figure 1 shows is, as already argued by us, that once all the canopy layers are taken into account then any apparent

multiple-modal patterns in the vegetation cover become much less apparent. But as we show below, ASS are not necessarily required to account for a fire-associated bimodal pattern in woody canopy cover in any case.

   To illustrate this point, we use the recently developed empirical parametrisation of fire effects on tropical ecosystems as presented by Veenendaal et al. (2016). In short, this model, developed using data from a range of "fire trials" allows the quantification of fire effects on canopy cover as a function of time of year and average fire frequency (or its reciprocal fire

return time) as is shown under pathway (a) of Fig 1. Here in this diagram - itself a simplified version of one given in Veenendaal et al. (2016), woody plant fractional cover ($\varphi_W$) in the absence of fire can be considered as being influenced solely by two factors: climate and soil (solid black lines in Fig. 1). Depending on the potential woody cover there is a certain allowable axylale (= grass + herb + forb) cover ($\phi_A$) – again based on Veenendaal et al. (2016) but here with some additional "noise" in the relationship (as detailed in the Supporting Information and as shown in Fig 1b). In terms of axylale/fire interactions we

consider two cases. Taking first case (1) we simulate a simple system where depending on $\phi_A$ – as indicated by the "switch" symbol, it is then considered that the vegetation formation in question can either potentially burn or not (Black step-function line in Fig 1c) with there thus being a critical $\phi_A$ below which fire cannot spread. In the second case (2: grey lines), it is assumed that - consistent with the data of Smit and Prins (2015) for example - that the lower the axylale cover the lower the fire frequency ($f$).

For the simulation (1) here we have taken the critical $\phi_A$ for the fire on/off switch as 0.2 and we also assume that in the "fire on" condition that fires can occur with equal probability across a range of different times of year (months given as for southern hemisphere) and average frequencies (ranging from burning every year to – on average – only every eight years), with the predicted effects shown for woody plant fractional covers in the absence of fire ($\varphi_0$) as predicted by the model of Veenendaal et al. (2016) for $\varphi_0$ of 0,.3, 0.5 and 0.7 shown in Fig 1a. Here $\varphi_0$ may be regarded as essentially the canopy cover expected for





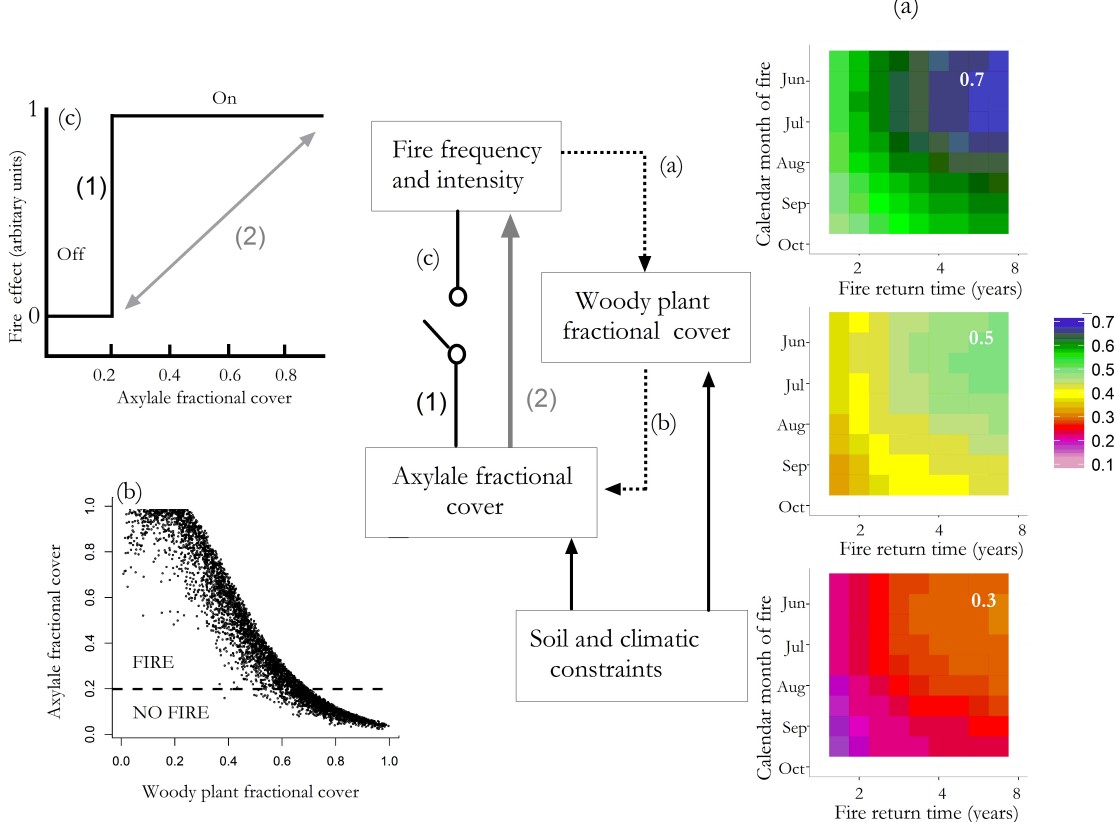

**Figure 1.** A simple mode of fire-vegetation feedbacks. As defined by Du Rietz (1936) we use the term "axylale" to unambiguously incorporate all grass-, herb-, sedge- and forb-type life forms into the one category without any necessary reference to taxonomic grouping (Ingrouille and Eddie, 2006; Torello-Raventos et al., 2013). For further description of the panels and their meaning, see the main text.

a "fire–protected control" with Fig 1a showing the well documented effects of fire frequency and time of year on canopy cover reductions (Veenendaal et al. 2016).

Taking an arbitrary pre-imposed distribution of $\varphi_0$ as shown in Fig. 2a, the resulting $\varphi_W$ for situation (1) are shown in Fig. 2b where, according to the definition of Torello-Raventos (2013), we have differentiated "savanna" from "forest" on the basis of the former having $\phi_A > 0.2$ (which for these simulations also has the convenient effect of having all savannas burn and all forests not). Here the simulated distribution with fire is indeed bimodal, but importantly, for each $\varphi_0$ there exists only one stable state $\varphi_W$. And moreover, with a simple switch mechanism operating, a long as $\phi_A > 0.2$ there is no feedback between axylale cover extent and fire frequency and/or the time of year that the fire occurs in these simulations.

But does that mean there are no fire-mediated feedbacks occurring at all in this simulation? Well, actually there is a feedback but it is more directly associated with canopy structure rather than fire. That is to say, in the simulation of (1) the tree cover affects axylale cover which then in turn affects the tree cover through either permitting fire to occur or not. In that sense we



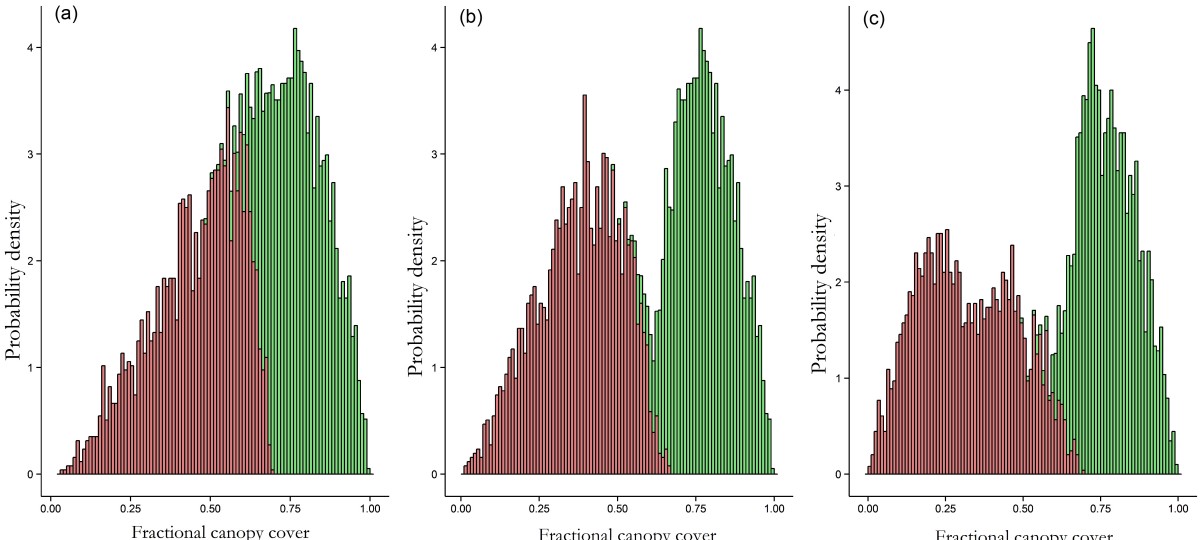

**Figure 2.** Predictions of the simple model of Fig. 1 for (a) no fire, (b) a grass-fire feedback as indicated by relationship (1) in Fig 1c and (c) a grass-fire feedback as indicated by relationship (2) in Fig 1c. In all cases have differentiated "savanna" (red bars) from "forest" (green bars) on the basis of the former having $\phi_A > 0.2$

would suggest that (1) reflects a "fire effected feedback" (as opposed to a fire "mediated" feedback"). Situation (1) might be considered analogous to the situation in dry tropical forests where, even in the absence of an axylale layer, leaf litter from the trees themselves contributes to a below canopy fuel load accumulating to a critical point late in the dry season (e.g. Miles et al 2006).

But what about situation (2)? Here with fire frequency increasing with increasing $\phi_A$ it might be expected that any simulation would "spin out of control" as any imposition of fire must lead to a decrease in $\varphi_W$ (Fig. 1a) which then gives rise to an increase in $\phi_A$ (Fig. 1b) which then increases $f$ (Fig. 1c), this then increasing the fire severity further reducing $\varphi_W$ (Fig. 1a) and so on. But it turns out that because the effects of any given fire regime on $\varphi_W$ are best considered as proportional rather than absolute effects (Veenendaal et al. 2016) the lesser absolute effects at lower $\varphi_W$ mean that for every $\varphi_0$ a unique fire-associated $\varphi_W$

exists even with $f$ correlating positively with $\phi_A$.

Using a simple iterative procedure to find the solution, the result from such a simulation is shown in Fig. 2c. Here we have used the simple parametrisation of $f = \phi_A$ (that is to say for $\phi_A = 1.0$ there is a simulated fire return time of one year and for $\phi_A = 0.2$ it is 5 years), with stands with $\phi_A < 0.2$ again having no fires. This shows that, as expected, the $\varphi_W$ distribution for fire affected savanna stands to be shifted towards the left (more severe fire effects) but with the bimodal nature of the statistical

distribution maintained. Compared with (1) simulation (2) does have a second feedback loop (the opening up of the canopy by fire increases grass cover which increases the frequency of fire) but even so there is only one unique solution and Fig. 2c



represents the distribution of unique stable states for the fire affected population of $\varphi_0$ of Fig 2a and with two positive feedbacks occurring.

Thus, contrary to the interpretations of Hirota et al. (2011), Staver et al. (2011a), Staal and Flores (2015) and others (for example Xu et al. (2015) and Dantas et al. (2016)) bi-modality in the distribution of any vegetation structure metric does not

necessarily mean that two alternative stable states exist. Rather, as in the above scheme which is essentially a "sharpening switch" *sensu* Wilson and Agnew (1992), such distributions may simply reflect a system where fire merely reinforces the vegetation patterns as determined by climate and/or soils.

Moreover, as we have already pointed out, there are alternative explanations for bimodal canopy cover distributions for which fire-effected feedbacks need not even be the prime mechanism. For example, Veenendaal et al. (2015) argued that as

a result of the "new niche creation" of a shaded understorey at or around the point of canopy closure (for which $\varphi_W \approx 0.6$) then a rapid increase in total stand-level woody plant cover should then ensue with a "filling up" of this newly created shaded understorey environment by suitably adapted woody species. That is to say, there is probably there is very little difference in the climatic/edaphic conditions necessary to support a stand of $\varphi_W = 0.65$ as opposed to $\varphi_W = 0.9$. And with fire-mediated feedbacks having nothing to do with this.

Around the point of canopy closure, inter-tree competition for light should also start to influence canopy architecture with substantial increases in mean tree heights also expected around $\varphi_W \approx 0.6$ (Veenendaal et al., 2015), a point at least conceded as a possible alternative explanation by Xu et al (2015)for the non-unimodal distribution of tree heights they take as evidence for the widespread existence of ASS in the temperate zone.

But in any case, the demonstration of Figs 1 and 2 that fire-effected bimodal distributions need not necessarily be associated

with ASS, taken in conjunction with alternative explanations for bimodal canopy cover and/or height distributions not even involving fire (Veenendaal et al., 2015), strongly suggests that support for the existence of ASS from these much touted "distributional analyses" do not even meet the "weak warrant" criterion of Reiss (2015). That is to say as piece of scientific "evidence" providing support for the existence of ASS they are more or less useless.

But now returning to the simulations of Figs 1 and 2, are they also consistent with the arguments presented in Veenendaal et

al. (2015)? Here, if we were to believe SaF then probably not, as they suggested that because we argued that fire frequencies may be higher in the savanna plots as an indirect consequence of intrinsically lower canopy cover, that this also means we are also of a belief that "there are no negative effects of fire on trees". But nothing could be further from the truth and as we actually stated as our ante-penultimate sentence (Our) "conclusions are, of course, not necessarily at odds with the notion that the frequency and magnitude of fire – both natural and anthropogenic – can substantially affect savanna vegetation structure."

Indeed, in this respect the suggestion of SaF that we argued otherwise suggests that SaF do not understand that fire can have effects on tropical vegetation without these effects being of sufficient magnitude to initiate the sort of positive feedbacks required for alternative stable states to exist (Staver et al., 2011b).

Moreover, many of the additional references cited as providing "evidence" for alternative stable states by SaF similarly lack foundation. For example, SaF suggest that fire exclusion experiments such as Moreira (2000) and Higgins (2007) provide

supporting evidence for ASS, but the precise rationale behind this statement is hard to ascertain. Of course, as would also





be predicted by our model presented above, if one removes fire from a normally burning savanna landscape then the canopy will develop and biomass increase and then, if around, fire intolerant species from areas such as gallery forests (which follow the river courses of all but the driest of the savanna lands ) may then invade. But that does not mean that the area where the fire protection experiment was undertaken was originally forest. Indeed careful analysis of one paper often cited as providing

evidence in this respect *viz.* Moreira (2000) actually shows that even after 30 years of fire protection, very open savanna types in the Brazilian Cerrado such as *campo sujo* (literally "dirty field") whilst thickening and increasing biomass still fail to attain a stature and/or canopy cover even approximating that of nearby denser *(cerradão)* woodlands exposed to fires on an annual basis. Thus, although there may have been a clear fire effect on vegetation structure noted in that study, what was not made clear is that the variations in vegetation physiognomies observed across the landscape in question can in no way be explained by different

fire regimes or by implication ASS. On the other hand (and as we discuss further in Sect. 4) there have been numerous studies showing that variations in soil properties are associated not only with variations in cerrado vegetation physiognomic form Goodland and Pollard, 1973; Lopes and Cox, 1977; Cole, 1960), but also that soils are important in accounting for variations in canopy structural characteristics across the savanna-forest transition itself (Cole, 1986; Lloyd et al., 2015; Veenendaal et al., 2015).

All the above cases of supposed evidence supporting the existence of ASS as cited by SaF thus constitute nothing more than excellent examples of a "fallacy of confirmation bias" (LasBossiere, 2013). That is to say, although the evidence can be interpreted as supporting ASS, it can just as easily be taken as supporting an alternative or even contradictory hypothesis.

## 3   Model simulations and the "fallacy of misplaced concreteness"

According to SaF, a second class of evidence for ASS comes from mathematical models which predict that under otherwise

identical conditions, it turns out that it is theoretically possible for at least two different woody/axylale cover combinations to exist (Staver et al., 2011b; Van Nes et al., 2014; Baudena et al., 2015; Staal et al., 2015). Such a result requires a non-linear model that is dynamic (i.e. including processes such as growth and mortality) rather than the "static approach" we have taken in Fig 1. But with the mathematical properties of any such model giving rise to ASS long-understood and easily interpreted via simple matrix algebra concepts (May 1977; May 2001; Soeteart and Herman, 2010). Thus, that anyone so inclined can develop

a model that predicts the presence of ASS in forest/savanna systems (or elsewhere for that matter) is in no way surprising. And that just because it is that such models can be constructed, this in no way proves (or even supports) the existence of ASS. It just shows (like $2 + 2 < 10$) that ASS are mathematically possible.

This a clear case of Whitehead's "fallacy of misplaced concreteness" (Whithead, 1925). That is to say, models can never constitute reality. And they should never be taken as evidence (equivocal or other wise) for the existence of any particular

mechanism. An interesting example of this comes from the stomatal physiology literature where there are several competing models characterised by quite different conceptualised feedback processes, but that all (broadly speaking) predict the same environmental responses (Buckley and Mott, 2003). This family of models including one by Cowan (1995) that through a brilliantly conceived mechanism, solves the conundrum of how it is that one can observe reversible feed-forward stomatal





response response to humidity. But unfortunately the phenomenon so successfully predicted by that elegant model is these days considered to probably not even exist, having most likely been a measurement protocol artefact (e.g., Franks et al., 1997).

## 4    Soil vegetation associations and the "fallacy of suppressed evidence"

A third source of "evidence" cited by SaF is that of "vegetation mosaics observed in the field" by which we take to mean that
there have recently been a few studies that have failed to explain observed vegetation distribution on the basis of the measured climatic and/or soil properties (as cited by SaF these were Warman and Moles (2009), Favier et al., (2012), Hoffmann et al., 2012; Dantas et al., 2013 and Grey and Bond, 2015). Yet one of these papers in support of this viewpoint is a review that considered this question only in passing (Hoffmann et al., 2012) with another one virtually data-free (Warman and Moles, 2009) and with the study of Favier et al., (2012) not actually measuring the soil properties of the studied stands at all: but
rather inferring them from large scale soil maps in no way designed to allow inferences to be made at anything other than the continental scale - for a more detailed discussion of this point see Lloyd et al (2015). Moreover, the cited study of Dantas et al. (2013) actually found forest soils to be more fertile than those of savanna in similar climatic regions (as we suggest to be generally the case) and so it is not clear why SaF included it as part of that argument. Although perhaps the supposed support for ASS from Dantas et al. (2013) as perceived by SaF was simply - as argued by the original authors themselves -
that because across the savanna-forest interface there are large contrasts in plant traits (as would be expected for any transition from flammable savanna to non-flammable forest) then this is evidence for ASS; the apparent logic behind which totally defies us. Likewise, working in South Africa, Charles-Dominique et al. (2015) argue with no clear rationale that because their three different structural formation categories ("forest", "savanna" and "thicket") have distinct virtually non-overlapping species compositions that this somehow provides evidence for these three distinct vegetation types representing ASS.

Again we reiterate here that under uniform climatic conditions all that is required for there to be a mosaic is for there to be some differences in soil physical and/or chemical characteristics which, especially in transition zones, means that (in one case) the overstorey canopy does not have sufficient resources to close, then allowing grasses to establish and fires to occur and with subsequent reductions in canopy cover and biomass ensuing (and the vegetation having fire-adapted traits). But where edaphic conditions are sufficient for the relatively high canopy cover to effectively prevent the establishment of grasses, then (with
more shrubs in the understorey) a very different vegetation type ensues. Our proposed mechanism is consistent with the study of Lloyd et al. (2015) who found that otherwise enigmatic variations in canopy structure across a range of forest and savanna sites in South America were explicable on the basis of a simple model incorporating soil water and soil potassium availability. That study also showed that "stunted forests", similar in structure to the thicket of Charles-Dominique et al. (2015), tend to be associated with shallow high cation status soils and with a distinct species composition – see also Torello-Raventos et al.
30    (2013).

So, what then to make of the one remaining study cited by SaF *viz* Gray and Bond (2015) who working in a savanna/forest mosaic in South Africa concluded that despite forests generally being found occurring on more fertile soils than savanna, that this could not be the explanation for their locations within the landscape mosaic: the apparent reason being because "nutrient



stocks in the savanna soils exceeded threshold requirements". But this requires that any soil effect on vegetation be mediated solely by the nutrient requirements of standing stocks, and this is just one of several means by which soil effects on vegetation can be manifest (Lloyd et al. 2015). So in short, as for many other studies of a similar vein across a range of scales (e.g. Favier et al.., 2012; Lehmann et al., 2014; Dantas et al., 2016) that study shows, contrary to what is claimed by the authors, nothing of

substance as regards the presence or absence of ASS. This is because nothing like the the full range of likely possibilities for the underlying causes of the variations in vegetation cover being investigated were actually considered. It thus seems that by focusing on what is a conceptually attractive theory many studies actually end up ignoring important alternative explanations.

The recent study of Dantas et al. (2016) is an especially good example. As background: analysing data from a wide range of sources across Africa and South America, they argue that they had found evidence for ASS on the basis of two sets of

observations. First that this ground-based observational dataset was multi-modal and second, that because any given value of an empirically developed "resource index" ended up predicting systematically lower stand-level woody basal areas for vegetation formations classified as being in the "wooded grassland state" as compared to those in the "savanna state" which were in turn found to be predicted to have lower stand-level basal areas at the same "resource index" as "forest". Yet the obvious question, which Dantas et al. (2016) seem not have even considered, is whether their resource index included the right soil parameters

or not. This is especially the case as the Dantas et al. (2016) "resource index" only included pH (whilst generally speaking a good indicator of soil nutrient status, hardly a direct measure) and Cation Exchange Capacity (CEC), the latter because it is measured usually at pH = 7 bearing virtually no relation to the Effective Cation Exchange Capacity (ECEC) in the variable charge soils that dominate much of the tropical region (Sanchez, 1976; Buol et a., 2011 ). And with the ECEC in any case of being of limited value as that measure includes not only the the sum of "base cations" (*viz* SB = calcium, magnesium,

potassium and sodium) as extracted from the soil cation exchange complexes, but also any (potentially toxic) aluminium ions present. In some highly weathered tropical soils the latter may amount to more than 99% of the total CEC (Quesada et al., 2010, Quesada et al., 2011).

Making matters worse, rather than actually using real data as measured in the plots whose basal area they were trying to assess, Dantas et al. (2016) took best estimate values from an interpolated 1 km grid soil information system (Hengl, et al.,

2014), giving little consideration as to the possibility that the estimated values are also documented to come with substantial errors. As an illustration, Fig. 3 shows in the upper panel (ranked from left to right according to total base cation concentration) measured soil cation data (0– 0.3 m) from about 40 forest and savanna plots worldwide for which we have ECEC and CEC data available (and for which we also note in passing that there is a pan-tropical and cross-biome relationship of between SB and canopy covers: Veenendaal et al., 2015 ). The lower panel shows the associated measured CEC as well as best estimates from

the Hengl, et al. (2014) database for the 1 km grid squares associated with the actual plot location along with the associated 90 % upper and lower confidence intervals. Overall, Fig. 3 illustrates three important points:

1. Especially in soils of a low SB the ECEC may be dominated by aluminium. Thus ECEC should never be thought of as a surrogate measure of soil base cation status (or soil fertility)



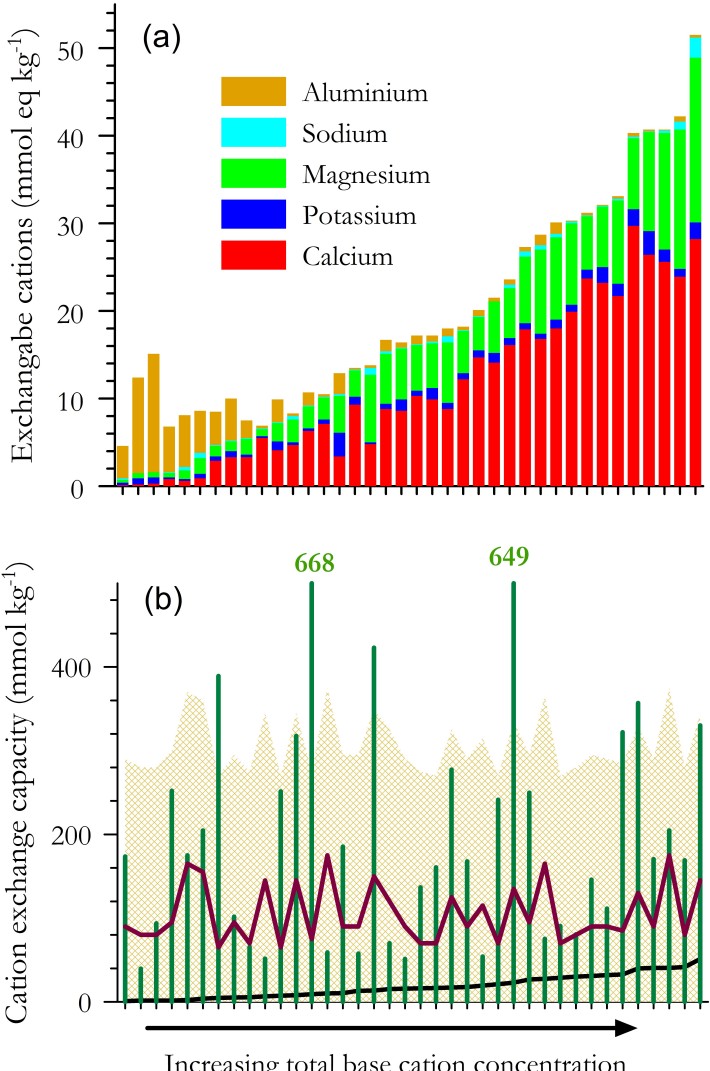

**Figure 3.** Marked differences in four contrasting metrics for expressing of the exchangeable cation status of 39 tropical soils.(a) exchangeable calcium, magnesium, potassium, sodium and aluminium concentrations (0–0.3 m depth) with the soils raked according to their sum of bases (SB). (b) measured cation capacity (CEC) for the same soils (green bars) along with best estimates of CEC for the same sites from the SoilsGrid1km database (Hengl et al., 2014) shown as a brown line and with the associated light brown shaded area indicating the quoted 90% confidence intervals for the SoilsGrid1km data. Emphasising the difference in scale, also shown repeated in (b) are the measured SB values (lower black line). Data represent forest and savanna sites sampled in Australia, Bolivia, Burkina Faso, Cameroon, Ghana and Mali as part the TROBIT project (Torello-Raventos et al., 2013: No such data was available for Brazil). Methodology as in Quesada et al. (2010) and Quesada et al. (2011).



2. Although CEC as usually measured at pH 7 may be an excellent and simple measure of soil clay activity and for that reason being an invaluable metric incorporated into many soil classification protocols (Buol et al., 2011), because both organic matter and variable charge clays have a cation exchange capacity that is pH dependent (Uehara and Gilman, 1981) in highly acid pH soils the ECEC at soil pH may be less than 5% of the CEC at pH7 and with little or no correlation between CEC and ECEC (for the dataset in Fig. 3, Kendall's $\tau = 0..09; p = 0.43$).

3. As cautioned by the manufacturers (Hengl et al., 2014) and can be seen from the confidence intervals in Fig 3b, all soil databases must inevitably be of a limited thematic and spatial accuracy. Indeed, as is given as a health warning for the very database used by Dantas et al. (2016) when talking about the interpolated data " ....suitability/usability for regional and global environmental models may be limited" see http://www.isric.org/content/disclaimer-soilgrids (accessed 17/11/2015).

In all the above examples, the authors involved did not explore the full range of climatic and/or edaphic drivers likely to be able to account for the observed vegetation distributions (with in several cases the soil variables not even measured!). We therefore classify all these studies as presenting cases of the "fallacy of suppressed evidence". Although the evidence may not necessarily have been actively suppressed, the error and remains the fallacy categorisation is applicable (Dowden, 2015).

Here as an aside we also note that given that hill-slope topography is the dominant landform in the humid tropics (Thomas, 1994) which through its spatial complexity confers a major source of soil variability as a consequence of differential rates of weathering of minerals leaching and lateral surface and subsurface transport processes according to landscape position (Moormann and Kang, 1978), then it is not at all surprising that vegetation relevant soil properties generally vary at a scale of order 100 metres (Moormann and Kang 1978; Coventry 1982; Stoop 1987; Bravard and Righi 1989; Davis et al. 1995; Dobermann et al. 1995; Dubroeucq and Volkoff 1998; Tchienkoua and Zech 2004; Tsui et al. 2004; Okae-Anti and Ogoe 2006). Thus, with even the more intensive soil surveys of the tropical regions being at a density around 1 profile per 1000 km$^2$ (Cooper et al., 2005) this means that irrespective of the sophistication of statistical interpolation technique, reliable non-directly measured soils data suitable for plot-level functional vegetation modelling data is all but an impossibility. Although the development of high-resolution remote sensing techniques in the future (e.g., Mulder et al., 2011) may provide one way forward.

## 5 Vegetation effects on soil properties and the "fallacy of wishful thinking"

Although apparently conceding that soil nutrient status may be important in influencing tropical vegetation structure and function – with forests typically of a higher soil nutrient status than savanna (albeit with one of the cited references in support of that statement coming from woodland/forest formations on the distinctly non-tropical island of Tasmania! ) – SaF rationalise their way around this in their Fig. 2 by supposing that this difference in soil fertility is due to

1. regular fires in savanna systems depleting the soil nutrients; and

2. forest species somehow enriching the soil, this then creating a positive feedback





In terms of the detrimental effects of fire on soil nutrients, we start our rebuttal by pointing out not only that the main conclusion of the reference cited by SaF in support of their thesis, *viz* Certini (2005) is actually that the effects of moderate intensity fires on soil nutrient status are actually positive and that, even when fires are severe, then as long as natural regeneration occurs then the negative effects of fire are much less than one might naively suspect. And with that paper also making the

obvious point – as long realised by boreal forest fire ecologists for example (Wirth 2005) – that there is a fundamental difference between surface fires (such as typically occur in savannas) as opposed to crown replacing fires in forests, the latter being the main subject of the Certini (2005) review. In particular, although in intense crown replacing fires soil temperatures at 50 mm depth may reach as much as 150°C (DeBano et al., 2000), for the relatively benign grass fires that characterise savannas, changes in soil temperatures are remarkably modest (Miranda et al., 2003). Of course, as pointed out by SaF there are some

nutrient losses during the consumption of (generally herbaceous) biomass by the passing fire fronts. But the question then is what happens to those nutrients? The answer is of course that - (because they cannot go up in a puff of logic or be transported to Mars) - that they return in the form of wet or dry deposition. This will not be at exactly the same location of course, but nevertheless most studies do show that indeed, for fire affected savannas that the nutrient inputs more or less balance the losses. For example, Pivello et al. (1992) estimated that for Brazilian cerrado the loss of nutrients associated with a typical fire would

typically be renewed via atmospheric inputs over a period of three to five years. This is pretty close to an average fire return time for relatively open savanna systems (Veenendaal et al., 2016) and consistent with this are the results of Pivello et al. (2010) who found no effect of long-term fire regime on soil nutrient stocks for the Miranda et al. (2002) fire trials in central Brazil. Likewise, Sawadogo et al (2005) found no effects of prescribed early dry-season fires on the soil nutrient stocks of savanna woodlands in Burkina Faso subject to a range of different grazing pressures. Although there are some reports of annual late-

season fires affecting soil nutrients (Mills and Fey, 2004; Pantami et al., 2010), it is important to bear in mind that such high frequency, high severity treatments are necessarily imposed through human activity and with effects on vegetation structure and function far in excess of those observed for any sort of natural fire regime (Veenendaal et al., 2016).

Thus, in short the assertion of SaF that non-anthropogenically imposed burning should serve to reduce soil fertility - either as as a generalisation or as applied specifically to savanna systems - is not supported by observation. But worse – the prime

reference that SaF cited in support of that claim does not even say that in the first place!

The above analysis also raises a basic point that is when examining the nutrient or carbon balance of an ecosystem, that it is necessary to consider not only the inputs, but also the outputs (and vice versa). And it is here that the assertion of SaF that forest trees somehow enhance soil fertility is similarly flawed. For example, from the Paiva et al. (2015) study they cite in support of this notion, both SaF and the original authors seem to think that just because as one moves along a transect from

riverine forest to savanna (where the main driver in ecosystem flux variability is surely water) that just because rates of nutrient input via litterfall increase from savanna to forest (mostly due to higher rates of litterfall in the forest part of the transect, but also due to slightly higher concentrations of foliar nutrients in the litterfall of forest trees ) that this then means that forest trees somehow enrich the soil; ignoring the fact that almost all of the nutrients found in the foliage and/or litter had to come from the soil in the first place (!). Indeed, a second paper cited by SaF as presented as showing that "when forest trees expand their

trees have positive effects on the nutrient ability of forest soils" (*viz.* Silva et al. 2008) similarly shows nothing of the sort. All



it shows is that in the Brazilian Cerrado riverine forest soils are more fertile than their slightly upland savanna counterparts which is, exactly what one would expect with ground water nutrients an important component of nutrient input into the former (Markewitz et al., 2006). Indeed, although it is clear that there are some very interesting and well documented feedbacks that may occur in the soil-vegetation system (Ehrenfeld et al., 2005; Phillips, 2009), this has never been shown to be the case for

the tropical forest/savanna system: especially as has already been shown above, and contrary to popular belief, there is actually no evidence for fires in savanna vegetation types actually reducing plant nutrient availability.

One recent paper that does, however, purport to show positive effects of forest trees and also cited by SaF is that of Silva et al. (2013): They measured above ground plant and soil nutrient stocks for five different vegetation types of vastly different soil nutrient and hydrological status and then – after observing little relationship between soil fertility and leaf or wood nutrient

concentrations – for some reason or other decided that this was somehow indicative of "vegetation structure generating and reinforcing gradients of resource accumulation as a result a slow redistribution from deep to surface soils through organic matter deposition": a statement not supported by the data presented and made all the more remarkable by the fact that one of the sites studied (a "deciduous forest") was acknowledged by those authors as clearly only growing where it was because of its very high cation content.

That is not to say of course that we do not agree with the general notion of plant nutrient cycling serving to concentrate nutrients towards the soil surface as is especially the case for strongly weathered soils (Quesada et al., 2011). But with for such soils the active weathering zone usually being located at depths far beyond the reach of roots (Baillie, 1989; Hamdan and Bumham, 1996) the notion that tropical forest trees can somehow access actively weathering materials which savanna trees somehow cannot – and by this process enrich the surface layers : see also Gray and Bond (2015) and Cowling and Potts (2015)

– should be regarded as a highly contentious and unsubstantiated view at best. Likewise, although it is also clearly the case that there are intrinsic differences between forest and savanna trees in terms of the former typically having higher dry weight foliar nitrogen and potassium concentrations (Schrodt et al., 2015; Lloyd et al., 2015), all that means is that forest trees typically have higher rates of uptake of N and/or K than savanna trees and/or longer leaf lifetimes. Thus, one should simply consider the tendency of savanna trees to be located on lower K–status soils as a plant–vegetation association consistent with the well

established principle of fast–growing plants with a high nutrient requirement typically being found on more fertile soils and vice versa (slow growing plants with a low nutrient requirement on less fertile soils: Lambers and Poorter, 1992; Veenendaal et al., 1996; Aerts and Chapin, 2000).

It thus seems that in terms of both the supposed detrimental effects of fire on soil nutrient stocks and the claimed positive effects of forest tree species on soil nutrient status – both of which are fundamental to SaF's proposed mechanism of nutrient/ASS

interactions (their Fig. 2) we have nothing more than several repeated cases of the "fallacy of wishful thinking". Because – on balance – there is actually no evidence to support either of these contentions at all.





## 6  Islands of fire and the "fallacy of hasty generalisation"

We started off this paper with a very positive interpretation of the paper of Jackson (1968) which, given our general questioning of the fire-associated ASS concept above, may perhaps surprise some readers. But in fact, the data and associated arguments in Veenendaal et al. (2015) were clearly only in reference to the tropical forest/savanna transition. This is contrary to what is implied in SaF who it seems do not appreciate the profound differences between stand-replacing crown fires typical of many fire-adapted forest systems (such as studied by Jackson, 1968) as opposed to the relatively benign surface-fires that usually occur in understorey of savannas. In short, it is not appropriate to cite studies tending to confirm the presence of ASS in the former (Wood and Bowman, 2012; Fletcher et al., 2015) as evidence for the latter. This is especially the case for the Fletcher et al. (2015) study of temperate zone eucalypt-grassland transitions where there were eco-hydrological feedbacks not applicable to most forest-savanna transitions also postulated.

The above constitute examples of the "fallacy of hasty generalisation": That is to say, just because for one area of the planet there are observations that "lend support" (to ASS: Wood and Bowman, 2012) or that may be considered to "satisfy a number of criteria" (of ASS: Fletcher et al., 2015), this does not mean ASS are a widespread reality (see also Scheffer et al., 2015).

## 7  Conclusion

The verification of ASS is, almost by definition, a very difficult question, especially in the absence of manipulated experiments (Petriatis and Latham, 1999; Schröder et a., 2005; Odion et a., 2010). Nevertheless, it is clear that none the arguments presented by SaF and their colleagues in support of their assertion of the widespread existence of ASS in the tropical forest/savanna systems can be considered to hold much weight, especially when one considers - as shown here - that all that is required to account for bi-modal distributions in plant cover is some sort of fire-effected 'sharpening switch' as defined by Wilson and Agnew (1972). Here, as opposed to ASS- the requirements for which are a relatively specific set of feedback interactions (Soeteart and Herman, 2010 ) – all that is required is for differences in climate and soil properties to determine where one or the other vegetation type is found, with the differences between savanna and forest then amplified by a simple fire-mediated mechanism that sharpens the contrast - this also resulting in the dominance of fire-resistant species in one vegetation type but not the other.

With no fewer than five fallacious argumentation types presented by SaF and many of these also apparently endemic to the ASS school of thought, we suggest that there has been a widespread exaggeration of what actually constitutes supporting evidence for ASS – with this arising, at least in part, through a widespread ignorance of the role of soils as important modulators of vegetation structure and function. Indeed, in general, there seems to be a widespread lack of appreciation amongst terrestrial ecologists as to the scale and extent to which there occurs variability in the many vegetation-relevant physical and chemical properties of the soil substrate (e.g., Bond et al., 2003; Favier et al., 2012; Murphy and Bowman, 2012; Lehmann et al., 2014; Dantas et al., 2016)

Interestingly, this has not, of course, always the been case: with - up to the time of Darwin at least - the science of "Natural History" readily assimilating knowledge from both the earth and biological sciences into a single framework (Detrosier, 1834;



Coleman, 1971; Worster, 1994). Thus, soil-vegetation associations were pre-eminent in the thinking of most early botanists exploring the tropical regions (e.g., Pickering, 1830; Gabb, 1871; Lamb, 1924; Burtt et al., 1942; Trapnell, 1950; Smith, 1951). This early emphasis on edaphic effects on vegetation distributions -now all but forgotten - occurred probably, not only because the required soils information was ready to hand, but also because these early field workers were unable to make comparative

inferences as to the long-term weather patterns of the areas they were studying.

Today the situation is the opposite as anyone can readily access reasonably high-quality climatological data for anywhere on the planet without even leaving the office, but with analogous data sets on soil properties inevitably of a much lower fidelity, even for relatively intensely studied areas (Grunwald et al., 2011). This has led to a overemphasis in terms of perceived climate effects on plant distributions and – with the domain of ASS essentially lying within the unexplained climate component – we

suggest that currently accepted ideas as to the wide-spread occurrence of fire-mediated ASS in tropical ecosystems (especially in terms of the forest/savanna contrast) may end up being little more that a case of *il silico* driven reification. In that respect, the sooner the majority of investigations as the underlying causes for variations in vegetation distributions are done with the help of a pick and shovel (as opposed to a computer) the better.

## 8 Acknowledgements

We thank Arie Staal and Bernando Flores for their commentary on Veenendaal et al., (2015) which, if nothing else, has forced us to sharpen our senescing minds and put to paper some concerns we have had for some time. Cation data in Figure 3 come courtesy of Mr. Martin Gilpin (University of Leeds). Valuable comments on an earlier draft of the manuscript were provided by Colin Prentice and Michael Bird with editorial support provided by Shiela Lloyd.



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
