# Peer review of "Are fire mediated feedbacks burning out of control?"

_Biogeosciences, 2015_

## Referee Comment (RC1) · Anonymous Referee #1 · 29 Mar 2016

General comments

There is a lot of very interesting information in this article concerning the control of savanna – forest boundaries and the relevance of Alternative Stable State theory. Unfortunately, however, it is written is a manner that detracts from these interesting perspectives: in essence it is a polemical rebuttal to a related article by Staal and Flores (2015, doi:10.5194/bg-12-5563-2015). These latter authors wrote a short paper that questioned the interpretation of a data-rich paper that previously appeared in Biogeosciences Discussions – based on vegetation data set collected from plots in tropical savanna and forest in Australia, Africa and South America – led by Lloyd and Veenendaal (Veenendaal et al. 2015 doi:10.5194/bg-12-2927-2015).

Because of the style of writing and the structure of the argument, Lloyd and Veenendaal's paper should not be treated as a stand-alone scientific contribution, rather it should be considered as another instalment in correspondence stemming from the paper by Veenendaal et al. (2015) in Biogeosciences Discussions (doi:10.5194/bg-12-2927-2015).

Specific comments

This paper is essentially an trenchant defence of two previous data papers (Veenendaal et al. 2015 doi:10.5194/bg-12-2927-2015 and Torello-Raventos et al. 2013 Plant Ecology Diversity, 6, 101–137, 2013). Lloyd and Veenendaal's modus operandi is to attack others workers, particularly Staal and Flores (2015, doi:10.5194/bg-12-5563-2015), who disagree with their conclusion that forest – savanna boundaries are controlled by edaphic factors, and that landscape fire merely sharpens such boundaries rather than creating them.

Lloyd and Veenendaal use a mix of rhetorical gestures, such as a reliance on philosophy, some quantitative modelling (the details of which are sketched being based on an unpublished paper Veenendaal et al. New Phytologist (submitted), reanalysis of existing soil data sets, and interrogation of the literature (which is far from comprehensive or even-handed).

All of these elements are framed around what the authors consider previous researcher's 'Fallacies'. There is some irony here as the current contribution by Lloyd and Veenendaal, the data paper by Veenendaal et al. (2015) and some others other from this group such as Lloyd et al. (2015, doi:10.5194/bg-12-6529-2015; Torello-Raventos et al. (2013) Plant Ecology Diversity 6, 101–137, 2013) suffer from similar methodological, logical and philosophical problems. These problems include reliance on simplistic computer models (Veenendaal et al. in a paper submitted New Phytologist so it is impossible to critically evaluate the logic), correlation being advanced as causation (Lloyd et al. 2015), very selective use of data and citations to promote a specific argument, poor experimental design relative to task in hand, omission of key data such

as any measurements of landscape fire and past disturbance history, and so on. Lloyd and Veenendaal ignore great swathes of research into forest-savanna boundaries.

Some of these problems are understandable and excusable because the problem of understanding the controls of tropical savanna and rainforest is extraordinarily complicated defying simple analysis or resolution, the fieldwork is logistically and physically demanding, and the relevant literature diffuse. Clearly, no single approach can solve this problem, and progress demands leveraging of existing research, targeted collection of more field data, collaboration among different disciplines, critical thinking and synthetic thought.

Most problematically, Lloyd and Veenendaal's contribution suffers from a lack of humility, there is a disturbing tone that 'I am right and you are all wrong'. I therefore suggest the authors move past their emotional response with the 'AAS community', and Staal and Flores (2015) in particular, and rewrite their argument in a more balanced and accessible style.

To reiterate, these authors have some extremely important data, deep insights and have the potential to make a landmark contribution in understanding the control of forest and savanna boundaries. To effectively communicate these strengths they need to rewrite this paper.

Technical corrections

The style of writing is verbose with numerous awkward, extremely long and convoluted sentences that lack commas. I recommend a careful rewrite to simplify the writing with shorter, and crisper sentences.

---

## Referee Comment (RC2) · Anonymous Referee #2 · 8 Apr 2016

General Comments

In this paper Lloyd and Veenendaal address several issues raised by Staal and Flores (2015) in response to a recent publication of Veenendaal et al. (2015), where Veenendaal and co-workers have argued that their field data were inconsistent with the hypothesis that tropical forest and savanna can be considered alternative stable states through a feedback between fire and low tree cover. In addition to refuting the arguments raised by Stall and Flores, Lloyd and Veeenendaal stated more clearly their argument that under uniform climate conditions, savanna-forest transitions would be more the result of differences in soil physical and chemical properties than the result of fire-vegetation feedbacks. Although they do not deny the importance of fire, they claim that fire-effected feedbacks simply serve to reinforce and sharpen the boundary between those two contrasting vegetation types, but they deny its role as determinant

factor governing the balance between humid tropical savanna and forest.

Interestingly, although they lay their argument considering both soil physical and chemical properties, they have a clear "bias" towards soil chemical properties. In this aspect both the alternative steady state (ASS) supporters and the edaphic–climate (EC) supporters agree that soil nutrient availability is a key-factor. In fact, there is such a tenuous difference in the arguments of both groups, that sometimes one thinks that they are both saying the same things, as I show in the following example. Hoffmann et al 2012a (Ecology Letters, (2012) 15: 759–768) claimed that soil characteristics per se cannot explain the persistence of savanna on well-drained clay soils that are widespread throughout the seasonal tropics. And quoting Hoffmann et al. (2012b; Austral Ecology (2012) 37, 634–643): "As fire can maintain open savanna conditions where climate and soils (here meaning enough soil nutrients) are otherwise able to support forest, fire-vegetation feedbacks permit the existence of alternate stable states". Lloyd and Veeenendaal disregard Hoffmann et al (2012a) statement as supportive for ASS by saying it was a review that considered this question only in passing. However, (and I quote Lloyd and Veeenendaal themselves) they restate that "under uniform climatic conditions all that is required for there to be a mosaic is for there to be some differences in soil physical and/or chemical characteristics which, especially in transition zones, means that the overstorey canopy does not have sufficient resources (meaning soil nutrients) to close, then allowing grasses to establish and fires to occur and with subsequent reductions in canopy cover and biomass ensuing (and the vegetation having fire-adapted traits). But where edaphic conditions are sufficient for the relatively high canopy cover to effectively prevent the establishment of grasses, then (with more shrubs in the understorey) a very different vegetation type ensues".

Are soil nutrients such a definitive determinant of vegetation structure in the Tropics? A clear example of the limitation of the soil nutrients-vegetation association can be seen in Figure 1 of Hoffmann et al. 2009 (Ecology, 90(5), 2009, pp. 1326–1337). As pointed by the authors, "forest soils had significantly greater pH, C, N, P, Ca, Mg, Mn, K, and

Zn and less available Fe and Cu than soils in adjacent savanna. However, there was considerable spatial variability in this overall trend, with two transects exhibiting little tendency for increased nutrient availability in the forest". I have no doubt that both EC and ASS supporters probably agree that local edaphic factors, such as shallow, sandy or seasonally flooded soils, might prevent some sites from ever becoming forest during fire suppression. However, these are not exceptions but the rule. Soil fertility and effective soil depth (as determined by the presence of concretions in the soil profile or nearness of seasonal or permanent water table to the soil surface) are both key determinants to govern changes in vegetation in tropical landscapes.

Local variations in vegetation physiognomy and floristic composition are more determined by soil properties and soil water regimes, particularly for savannas such as the ones of Central Brazil, where the presence of small watersheds throughout the biome provides the framework for an infinite variety of soil and vegetation mosaics. There is no single physiognomic type which covers the whole of a watershed in the cerrado landscape. Associated with variations in relief, ground water table level, drainage patterns and soils, the vegetation also changes. This is true not only for the tropical savannas but for the wet tropics as well. The mosaic of vegetation types in the Amazon region is another clear example of these complex interaction. Although in this paper the authors clearly stress the importance of soil properties in general, this was not true in their original paper (Veenendaal et al. 2015), where they were much more dogmatic and relied mostly on plant available soil water and soil cation status to explain observed changes in vegetation structure.

We should deal with the complexity of savanna and forest landscapes in the sampling designs and perhaps a watershed or landscape approach would be a more effective framework to understand past and predict future scenarios for tropical savanna and forests resulting from pressures imposed by changes in land use, fire regime and climate. Variations in relief, ground water table level, drainage patterns and soils have to be taken together in order to better understand patterns of distribution of savanna and

forests in tropical regions. In addition to vegetation structure, we should also incorporate species composition or at least functional groups to develop more realistic scenarios of changes in vegetation in response to anthropogenic global warming. Global and regional modeling efforts would also be more effective by taking into consideration the current fragmentation of tropical forest and savanna biomes.

In short, instead of just spending all this effort in deconstructing Stall and Flores arguments, Lloyd and Veeenendaal should move beyond and provide new insights to make this manuscript a novel contribution to our understanding of savanna-forest dynamics. Rebuttal, irony, deconstruction, self-confidence are not enough. I think readers would expect more from their "senescing" minds.

Specific comments

1. Perhaps I missed, but I could not find anywhere the range in rainfall. Are we discussing humid (wet; mean annual precipitation > 800 mm) savannas only?

2. The modeling approach adopted here by Lloyd and Veenendaal is still under review, and as such, not accessible (makes it hard to ascertain model assumptions and restrictions). In this way, it is still questionable at this stage and may change pending reviewer comments. However, as it was used more to illustrate the point that models cannot be used as "evidence" and that simulation results are strongly dependent on the model assumptions, it is perhaps acceptable and worthwhile to have it here.

3. It seems to me that their modeling approach requires uniform and steady state climatic conditions. Perhaps authors should state this more clearly.

4. Lloyd and Veenendaal argue that all references cited as providing "evidence" for alternative stable states by Staal and Flores lack foundation. This is perhaps true for the ones cited by by Staal and Flores, but there are several fine examples in the literature that this is happening. One of the most interesting that I know of it is the work of Pinheiro and Durigan (2009; Revista Brasil. Bot32, pp. 441-454) where they used

aerial photographs to show a change of open savanna physiognomies to a forest physiognomy with a continuous tree stratum in protected areas after a few decades. I raised this example to show the limitations of developing the whole argument based only on vegetation structure (plant cover). Whether this particular example would support one view or the other will depend on definition of what is a "forest" and what is a "savanna". Some would claim that in this particular case, the vegetation structure is changing towards a savanna woodland and others, to a dry or xeric forest. Or stating in other words, species composition does matter, not only cover or structure. Sadly, both the ASS and EC supporters have based their whole argument (at least in this exchange in Biogeosciences) on vegetation structure per se. Species composition or perhaps better differences in functional diversification between the two types of vegetation is not much considered, as well as the many other vegetation types that are all lumped together as either "savanna" or "forest".

5. The potential expansion of gallery forests into the savanna has also been demonstrated (for instance Silva et al. 2008; Global Change Biology 14, 2108–2118), using 14C analysis and Carbon isotope ratios of of soil organic matter and later confirmed by vegetation surveys of seedlings, juveniles and adult trees across the studied savanna-forest boundaries (Geiger et al. 2011; Journal of Vegetation Science 22 (2011) 312–321), highlighting the importance of forest-savanna interfaces. It is therefore reasonable to expect that forest tree establishment along borders allows the long-term persistence of forest patches, and promotes fast forest expansion under favorable climatic conditions.

6. On the other hand, it is true that this is not happening everywhere or for all savanna-forest interfaces. Reality is much more complex, as shown by Silva et al. (2010; Plant Soil 333:431–442) and using the same techniques, past vegetation changes or stability in Amazonian Savannas (Sanaiotti et al. 2002; Biotropica 34: 2-16. Similar patterns were also reported for Cameroon by Desjardins et al. (2013; Comptes Rendus Geoscience 345: 266–271. Pollen and charcoal records also point towards a complex

picture of savanna and forest temporal dynamics. In this regards, see for instance Ledru (2002; Late Quaternary history and evolution of the Cerrados as revealed by palynological records. In Oliveira and Marquis (eds), The Cerrados of Brazil: ecology and natural history of a neotropical savanna).

---

## Author Comment (AC1) · 13 Jun 2016

We thank Anonymous Referee 1 (R1) for his/her compliment to us on the information we brought forward to question the response by Staal and Flores (2015) a.k.a. SAF. (S)he does, however, take issue in particular with the style of writing of the paper and again in particular with the polemical writing style.

While realising that our direct approach may have upset some colleagues (for which EMV duly apologises) it has also led to discussions in many departments among scientists that are interested in the same issues at hand. We have had lively debates inside and outside conference rooms and email exchanges between ourselves and colleagues that we have criticised and new grants are being/have been applied for to work on formulating and testing hypotheses that emanate from (paraphrasing R2)"ASS

versus EC theory". In that sense we feel that we have reached our objective.

Responding to specific comments: We note R1's backhand criticism of our own studies that use correlation to put forward alternative explanations for tropical vegetation structure and its transitions. But that does not really hold up: For example, as regards Lloyd et al (2015), the very wording of the title of the paper reads: "Edaphic, structural and physiological contrasts across Amazon Basin forest–savanna ecotones SUGGEST a role for potassium as a key modulator". Here the "suggest" immediately alludes to the more nuanced interpretation of our dataset than has been the case for those putting forward things like bimodal distributions of remotely sensed canopy cover as unequivocal support for ASS (with that, of course, being one main focus of our critique). Also in Torello Raventos et al. (2013) we are almost entirely concerned with the use of numerical techniques to classify tropical vegetation types and so it is hard to see what the exact criticism of R1 is here. In Veenendaal et al. (2015) we use correlation and regression as descriptive statistics to show that there are trends in tropical vegetation structure which are associated with climatic and edaphic factors to, in the Discussion of these papers, simply raise the possibility that these provide alternative interpretations to ASS theory. So, this being in a public forum, for R1 to anonymously make the *non sequitur* assertion that we are guilty of similar "sins" to those we are criticising comes across as pretty much "on the nose". Likewise, our (unpublished model) was merely introduced to demonstrate that there are reasonable explanations for bi-modal distributions in vegetation cover that do not necessarily involve ASS: something the ASS community seem — up until now at least – to have not even considered as a remote possibility.

And of course, that we see a need for rigorous testing of our own hypothesis. But in responding to R1 the critical point here is that we have never presented our own ideas as anything other than that (i.e. theories to be rigorously tested), but that this to date has not been the case for the widely accepted hypotheses that it is ASS mediated by fire mediated feedbacks that dominate the observed variations in tropical vegetation

structure worldwide. Hence, being prompted by SAF, our finding the need to write our critique in the first place.

Anonymous Referee 2 (R2) provides a further detailed contribution to the debate. This review almost stands alone as an additional contribution to the discussion at hand and we wholly agree with many of his/her ideas as to the importance of e.g. soil water regime, local re-distribution of runoff water, ground water resources, flooding, effective soil depth etc. These are indeed essential when understanding pattern and process in the field, and with the response of tropical vegetation cover to these edaphic drivers almost certainly also climate specific. We also note here that for the sake of clarity our commentary did not attempt a full review of all factors determining Forest–Savanna boundary patterns. But rather focussed on SAF and some associated papers. In doing so we deliberately also chose a somewhat "trenchant" approach (R1) so as to get the debate more clearly focussed and to provoke discussions on how fire soil, edaphic factors, and climate (including carbon dioxide increases) may interact. The review of R2 provides a valuable further contribution to that discussion. In terms of the specifics of what we do and do not think, we do, however, point out that the suggestion of R2 that we consider chemical effects to be more important than physical effects when considering soil influences on tropical vegetation structure and function is not correct (see for example Discussion in Lloyd et al., 2015).

R2 also questions the difference between ASS and EC proponents - and (s)he certainly has a point here if one considers that when soil conditions improve (soil fertility, soil physical conditions) – then fire impacts may be expected to be reduced under closing canopy. However, a main difference in opinion is evident. For ASS the whole concept hinges on contrasting fire tolerant trees in open pyrogenic and fire sensitive trees in closed non–pyrogenic environments, with a threshold for fire activity at a certain cover (say $> 0.60$). The proposed mechanism also needs a relatively short return time of fire events being the norm and all this being applicable in the climate range above 1.1 m of rainfall where most Forest–Savanna transitions exist. It also requires stable states

with transitional vegetation forms being intrinsically unstable. ASS theory thus relies on no long term transitional formation type vegetation being possible: something very much at odds with our own observations. For example, there are in the humid tropical transition zones globally a wide differentiation in vegetation structure, as can indeed already be deducted from floristic literature (e.g. for cerrado/cerradão the work by Ratter and others; for West African forest and drier areas work by Swaine, Hawthorne, and others) and as implied in general for forest–savanna transitions in Torello Raventos et al (2013) with such "transitional dry forests and woodlands" in a relatively stable state under conditions determined by soil, climate, and with their own endogenous fire regimes.

More on that in later publications of course, as we fully intend in the future to develop these idea further: this most likely being in conjunction with future work on the interactions between climatic and edaphic factors in forest/savanna transitions world wide (and with the aid of a spade!).

**But our commentary on SAF was never intended to be a comprehensive review and in its final version we believe it should remain sufficiently focussed as such.**

Specific comments to R2.

1. Indeed we focus mostly on humid transitions (above 1 m annual rainfall) which include most forest transitions. Our model approach does indicate that (in a relative sense) fire may play a more important role on woody cover in drier regions as has also been reported by others.

2. We are hopeful that Veenendaal et al (2016) will be accepted soon (it is currently being revised having spent many months out in review).

3. We will do so although we note that unless severe human management is included (removal of woody canopy cover) not even severe droughts are likely sufficient to cause canopy damage of a magnitude sufficient to cause a permanent transition to a more open canopy type.

4. -7. We do not agree that fire exclusion studies in themselves are necessarily proof for fire as an agent of alternate stable states. This is because fire is an inherent component of open tropical vegetation with a dry season, but that in forest/savanna transitions its effect (in the absence of human activity) need not necessarily ever be large enough to cause a tipping point. We agree with other statements on effects of climate fluctuations and would also mention $CO_2$ increase here.

Finally we note (this being in relation to both R1 and R2) that neither referee actually raises any serious scientific concerns as to the validity of any of our arguments. Thus our questioning of some generally held ASS "truths" seems to have served a useful purpose. We also know that whilst our writing style has obviously being perceived by some as not showing a sufficient lack of humility, others have viewed our honesty of approach as more like refreshing (indeed to quote the 1930's US journalist/author Isaac Goldberg with a few modifications: "Diplomacy mostly serves to say the nastiest things, but in the nicest way"). We hope that in the end our non–diplomatic critique will serve to not only stimulate new debates as to the nature of forest/savanna transition zones, but also to help widen appreciation as to potential pitfalls when it comes to hypothesis testing and the interpretation of available "evidence".